# Safety and Protective Effects of Influenza Vaccination in Pregnant Women on Pregnancy and Birth Outcomes in Pune, India: A Cross-Sectional Study

**DOI:** 10.3390/vaccines11061034

**Published:** 2023-05-28

**Authors:** Hanif Shaikh, Pranesha Koli, Vaishali Undale, Anil Pardeshi, Mahesh Asalkar, Sushant Sahastrabuddhe, Anand Kawade, Chandrashekhar Upasani

**Affiliations:** 1Department of Pharmacology, SNJB’s Shriman Suresh Dada Jain College of Pharmacy, Chandwad 423101, India; 2International Vaccine Institute, SNU Research Park, Gwanak ro, Seoul 08826, Republic of Korea; 3Department of Pharmacology, Dr. D. Y. Patil Institute of Pharmaceutical Sciences and Research, Pune 411018, India; 4Department of Clinical Pharmacology, Seth G.S. Medical College, KEM Hospital, Mumbai 400012, India; 5Department of Obstetrics and Gynecology, Pimpri Chinchwad Municipal Corporation’s Postgraduate Institute, Yashwantrao Chavan Memorial Hospital, Pune 411018, India; 6Vadu Rural Health Program, KEM Hospital Research Centre, Pune 412216, India

**Keywords:** maternal immunization, influenza vaccination, pregnant women, pregnancy outcome

## Abstract

Background: Maternal influenza vaccination provides effective protection against influenza infections in pregnant women and their newborns. In India, the influenza vaccine has not yet been offered through immunization programs, owing to the lack of sufficient safety data for pregnant Indian women. Methods: This cross-sectional observational study enrolled 558 women admitted to the obstetrics ward of a civic hospital in Pune. Study-related information was obtained from the participants through hospital records and interviews using structured questionnaires. Univariate and multivariable analysis was used, and the chi-square test with adjusted odds ratio was estimated to account for vaccine exposure and the temporal nature of each outcome, respectively. Results: Women not vaccinated against influenza during pregnancy had a higher risk of delivering very LBW infants, and possible protective effects were suggested (AOR 2.29, 95% CI 1.03 to 5.58, *p* = 0.03). No association was observed between maternal influenza vaccination for Caesarean section (LSCS) (AOR 0.97, 95% CI, 0.78, 1.85), stillbirth (AOR 1.8, 95% CI 0.18, 24.64) and NICU admission (AOR, 0.87, 0.29 to 2.85), and congenital anomaly (AOR, 0.81, 0.10 to 3.87). Interpretation: These results show that the influenza vaccine administered during pregnancy is safe and might lower the risk of negative birth outcomes.

## 1. Introduction

Influenza infection causes substantial morbidity and mortality in pregnant women and young children. Immunization of pregnant women with an influenza vaccine is effective in reducing the risk of influenza and has been reported to be safe for mothers and their fetuses [1,2]. It also reduces the risk of influenza in infants during the first six months of life [3]. In October 2009, India witnessed an influenza (H1N1) pandemic [4]. Influenza is still endemic in some parts of country, including Pune, the city in the state of Maharashtra that was most severely affected during the pandemic [5]. In India, there is distinct seasonality across different cities that might be related to latitude and environmental factors. A study carried out to understand the dynamics of influenza seasonality in some of the cities in India showed that influenza circulation is primarily from June to October with discrete peaks in July–September coinciding with monsoons in cities such as Delhi and Lucknow (north), Pune (west), Allaphuza (southwest), Nagpur (central), Kolkata (east), and Dibrugarh (northeast), whereas Chennai and Vellore (southeast) revealed peaks in October–November, coinciding with the monsoon months in these cities [6]. In 2012, the World Health Organization (WHO) Strategic Advisory Group of Experts (SAGE) recommended that any country with an influenza immunization program should prioritize pregnant women [7]. Following the WHO recommendation and considering the endemic situation and availability of global data on influenza vaccine effectiveness, in 2015, the Maharashtra state government instructed local health care authorities of endemic regions to vaccinate high-risk populations, such as pregnant women, with a seasonal inactivated influenza vaccine free of charge. Since then, a few selected civic hospitals in Pune have been vaccinating pregnant women with the Trivalent Inactivated Influenza Vaccine (TIV3). However, since the beginning of the vaccination drive, we observed inadequate promotion by health care providers. Additionally, there was an interrupted supply of influenza vaccine, resulting in poor coverage in the Pune area. Although there are no major safety concerns reported for maternal influenza vaccines across the region, opinions on the effect of influenza vaccination on adverse birth outcomes vary [8,9,10].

Observational studies conducted in Canada and Australia did not find an association between influenza vaccination during pregnancy and adverse fetal or perinatal outcomes [11,12]; however, another study conducted by Donahue et al. reported a possible association between influenza vaccination administered very early in the first trimester and spontaneous abortion [13]. A review of previous studies on the safety of influenza immunization among pregnant women revealed that no studies have examined the influence of influenza vaccination on pregnancy and neonatal outcomes in pregnant Indian women. This lack of evidence among pregnant women may present a barrier to endorsing the influenza vaccination for this population. In addition, the effect of vaccination may vary by population- or geographic-specific factors, such as influenza seasonality and baseline rates of low birth weight or preterm births. Therefore, this study aimed to examine vaccine safety by comparing the incidence of adverse pregnancy and birth outcomes between vaccinated and unvaccinated pregnant women in Pune.

## 2. Materials and Methods

### 2.1. Study Design, Setting, and Study Population

This cross-sectional, observational study was conducted at the Yashwantrao Chavan Memorial Hospital (YCMH), a tertiary care civic hospital in Pune that has a good facility for managing high-risk pregnancies with a neonatal ICU. Pregnant women attending Antenatal Care (ANC) were offered a Trivalent inactivated Influenza Vaccine (TIV) by the hospital. Vaccination was recommended mainly during the second or third trimester of pregnancy, although we noticed that a few pregnant women had received the vaccine in the first trimester of their pregnancy. It was also noted that most women receiving the vaccine in the YCMH preferred to deliver in the obstetrics ward of the same hospital. We approached women who delivered in the YCMH and were admitted to the obstetric ward between October 2019 and March 2020. These women were approached within 48 h. of their admission in the obstetric ward and were asked for their participation in the study. Voluntarily provided written informed consent was obtained from each participant to get access for their records and to interview them to check their socioeconomic status, medical history, past pregnancy status, alcohol consumption, and tobacco consumption using a structured questionnaire. Maternal characteristics were collected from all the participants.

The study was reviewed and approved by the Institutional Ethics Committee of KEM Hospital Research Centre Pune. We followed the ICMR’s Ethical Guidelines for Biomedical and Health Research on Human Participants (2017) [14].

### 2.2. Exposure

Influenza vaccination during pregnancy was defined as a vaccine received between the first day (date) of the last menstrual period and the end of pregnancy. It was an exposure of interest for this study. The vaccination status of the participants was ascertained from the influenza vaccination stamp on their Antenatal Care (ANC) card. The vaccination date and gestational weeks at the time of vaccine administration were recorded. 

### 2.3. Outcomes

Pregnancy-related adverse events/complications such as pregnancy-induced hypertension, gestational diabetes, preeclampsia, chorioamnionitis, premature preterm rupture of membranes (PPROM), spontaneous abortion (occurring before 20 weeks of gestation) and/or preterm birth, and birth outcomes including congenital anomalies, low birth weight (<2500 g) (LBW), very low birth weight (<1500 g), Apgar score at 5 min, NICU or neonatal care unit hospitalization, respiratory distress syndrome, and mechanical ventilation were assessed. The Brighton Collaboration guidelines were used to diagnose pregnancy and birth complications [15]. The expected date of delivery was calculated from the last menstrual period (LMP) date.

### 2.4. Covariates

Information regarding sociodemographics, age, education, smoking, medical and obstetric history, complications during pregnancy, and vaccination status for the influenza vaccine was collected using hospital records and through a questionnaire-based interview. 

### 2.5. Statistical Methods for Analysis

The demographic and clinical characteristics of the study participants were descriptively analyzed. Counts (*n*) and percentages (%) were used to describe categorical variables. Continuous variables were estimated as mean ± standard deviation (SD). Where distribution was not normal, median and IQR were used. To evaluate whether there was an association between influenza vaccination status and each outcome variable, chi-squared tests of association were applied to the variables. The intergroup statistical comparison of the distribution of categorical variables was performed using the chi-square test or Fisher’s exact probability test if more than 20% had an expected frequency of less than 5.

The intergroup statistical comparison of the distribution of means of continuous variables was performed using an independent sample *t*-test. For all multivariable models, different levels of education, maternal health risk factors (such as age, gravidity, smoking, and alcohol intake), pre-existing health conditions, history of surgical operations, or abortions were among the variables selected as potential confounders based on the available literature [16,17]. Multivariable logistic regression analysis was performed to obtain statistically significant and independent determinants of the incidence of abnormal outcome measures (pregnancy and neonatal outcomes). The underlying normality assumption was tested before subjecting the continuous variables to a *t*-test. 

In the entire study, *p*-values less than 0.05 were considered statistically significant. All data were statistically analyzed using the Statistical Package for Social Sciences (SPSS version 24.0, IBM Corporation, Armonk City, NY, USA) for MS Windows.

## 3. Results

We approached 595 eligible women admitted to the obstetrics ward. Thirty women refused to provide informed consent. Written informed consent was obtained from 565 women, and they were further screened. Of the 565 women screened, two withdrew consent during screening, and three participants had no mother and child protection card or medical notes; hence, they were excluded from our final analyses. Two women who had received influenza vaccination within six months prior to their pregnancy were excluded. The final cohort comprised 558 women (Figure 1). 

Of the 558 women recruited from the obstetric ward, 265 (47.5%) received an influenza vaccine during their pregnancy. Maternal characteristics did not differ significantly between vaccinated and unvaccinated women. The mean maternal age of the participants was 24.7 years (SD 4.4) (range: 15–38 years), and the median gestational age at influenza vaccination was 28 weeks (IQR 12–38 weeks). Higher educational level was found to be significantly associated with vaccine acceptance (*p* < 0.001) (Table 1).

### 3.1. Outcomes

The overall uptake of influenza vaccination was 47.5% (265/558); of these, 1.9% (*n* = 5) were vaccinated in the first trimester, 40.4% (*n* = 107) in the second trimester, and 57.7% (*n* = 153) in the third trimester (Figure 2a,b). A total of 77.8% of uneducated women in the study (*n* = 84/108) did not receive influenza vaccination during pregnancy.

#### 3.1.1. Pregnancy Outcomes

Of the 558 study participants enrolled, 3 participants had stillbirths (0.53%), and 555 (99.47%) participants delivered a live infant. We observed that 43 (7.7%) participants had a history of one or more spontaneous abortions. We did not observe cases of spontaneous abortion admitted in the obstetrics ward as most of the women and relatives preferred not to remain hospitalized in the ward after abortion. There was no statistical difference in stillbirth rates between vaccinated (*n* = 1) and unvaccinated women (*n* = 2). Our time-dependent analysis showed no association between influenza vaccination through 37 weeks of gestation and premature rupture of the membranes. Our model showed that influenza vaccination during pregnancy was not associated with gestational hypertension. 

Overall, 9.3% (52 of 558) of pregnancies resulted in preterm births. The mean gestational age at delivery was 37.8 weeks (SD 2.4 weeks), and time-dependent analysis showed no association between influenza vaccination through 37 weeks of gestation and preterm birth.

#### 3.1.2. Birth Outcomes 

Influenza vaccination during pregnancy had no effect on the risk of congenital anomalies (*p* = 0.672). There was no association between maternal influenza vaccination and adverse neonatal outcomes, including Apgar scores at 5 min (adjusted odds ratio [AOR] 0.92, 95% CI 0.23 to 3.78), neonatal care unit (NICU) hospitalization (AOR 0.87, 95% CI, 0.29 to 2.85), mechanical ventilation (AOR 0.72, 95% CI, 0.21 to 4.00), or respiratory distress syndrome (AOR 1.1, 95% CI, 0.35 to 3.95). 

#### 3.1.3. Birth Weight Variable 

Birth weight (g is a continuous numerical variable compared across vaccination status modalities. The distribution of birth weight (g) was not normal among the different groups. The mean weight of infants born to vaccinated mothers was 2693 ± 503 g, and for infants born to unvaccinated mothers it was 2579 ± 602 g. In the present study, the mean birth weight was found to be 2633 g (Table 2). There was no evidence of an increased risk of LBW associated with the receipt of inactivated influenza vaccine during any trimester of pregnancy. Women not vaccinated against influenza during pregnancy had a higher risk of delivering very LBW infants in our univariate and multivariable analyses [AOR] 2.29, 95% CI; 1.03 to 5.58, *p* = 0.03).

Preterm: babies born before 37 weeks of pregnancy; full-term: 39 0/7 weeks through 40 6/7 weeks; post-term: 42 0/7 and beyond. LSCS: lower (uterine) segment Caesarean section. LBW: weight at birth of <2500 g; VLBW: weight at birth <1500 g. High-risk pregnancy: a pregnancy complicated by disease or disorder that may endanger the life or affect the health of mother, fetus, or newborn, and same was documented in patient record.

As shown in Table 2, the distribution of gestational age differed significantly between the nonvaccinated and vaccinated groups (*p*-value < 0.05). Multivariable analysis did not show any association between vaccination status and prematurity and birth weight, but in univariate statistical analysis the distribution of outcome measures, such as VLBW, was significantly associated with vaccination status (*p*-value < 0.05).

On multivariable statistical analysis, the distribution of outcome measures such as pregnancy outcome (stillbirth), preterm gestation, LBW, and a lower (uterine) segment Caesarean section (LSCS) mode of delivery was not significantly associated with vaccination status after adjusting for confounders such as maternal age, education, and gravidity (*p*-value > 0.05). On multivariable statistical analysis, the distribution of outcome measures such as VLBW was significantly associated with vaccination status (*p* < 0.05). 

It was also shown that pregnancy outcomes such as the occurrence of high-risk pregnancy, occurrence of complications before or during delivery, and occurrence of complications after delivery were not significantly associated with vaccination status (*p*-value > 0.05).

As mentioned in Table 3, On multivariable statistical analysis, pregnancy outcome, the occurrence of high-risk pregnancy (where mother or fetus has an increased risk of adverse outcome), occurrence of complications before or during delivery, and occurrence of complications after delivery, and neonatal outcome, the distribution of the incidence of various neonatal complications such as abnormal Apgar score at 5 min, NICU requirement, requirement of mechanical ventilation, occurrence of respiratory distress, occurrence of fetal distress, and occurrence of congenital anomaly, were not significantly associated with vaccination status.

## 4. Discussion

Reports published earlier showed that the influenza infection during pregnancy was associated with an increased risk of adverse pregnancy outcomes, including preterm births, stillbirths, low birth weight, and miscarriage [18,19,20,21]. Antenatal influenza vaccination is the best strategy to avoid both morbidity and mortality among pregnant women and infants; however, there are always concerns about the safety of influenza vaccines in terms of their effects on pregnancy and birth outcomes. 

The findings of our study, conducted in an Indian setting, suggest that the influenza vaccination during pregnancy was not associated with adverse birth outcomes. This result is consistent with randomized controlled trials conducted in other countries, such as South Africa and Nepal [1,22]. Several other studies and meta-analyses also demonstrated that antenatal vaccination with seasonal inactivated influenza vaccines does not increase the risk of fetal death, adverse fetal and birth outcomes, low birth weight, spontaneous abortion, or congenital malformations [17,18,23,24]. Our observation regarding the improvement in birth weight among neonates in the vaccinated group is similar to that of a randomized controlled trial conducted in Bangladesh, which showed that maternal influenza vaccination during pregnancy was associated with an increase in the mean birth weight of babies born during the influenza season [25]. Interestingly, a pooled analysis of randomized control trials conducted in Nepal, Mali, and South Africa showed no overall association between maternal vaccination and low birth weight [22]. Therefore, the lack of a biologically plausible mechanism for the suggested association raises concerns for any interpretation of positive correlations between vaccination and an increase in mean birth weight among infants. The improvement in health conditions in the absence of influenza and other cofactors might have improved the birth weights of the neonates of vaccinated mothers. The finding of lower VLBW in the vaccinated group also may be explained just by chance.

A case–control study conducted by Donahue et al. reported an association between spontaneous abortion and vaccination with inactivated influenza vaccine within a 28-day exposure period, but only among women who had received an A(H1N1)-containing vaccine in the previous influenza season [13]; however, this study failed to gain confidence due to the difficulty in interpreting the data.

Another study conducted by Eick et al. indicated a significant association between influenza vaccination in pregnant women and a reduced risk of influenza virus infection or hospitalization for influenza-like illnesses up to six months of age [26]. In our study we could not collect enough information on cases of influenza occurred among pregnant women during pregnancy.

### Strengths and Limitations of the Study

Our study had several strengths. The pregnancy outcome data were directly received from the delivery records, and the information captured from the participants through a questionnaire for medical history was also confirmed from the physician’s notes and hospital records simultaneously, which improved the quality of the data. Vaccinated and nonvaccinated participants were exposed to the same season and had similar seasonal influenza effects. 

The limitations include that comparatively few participants were recruited because of the interrupted supply of influenza vaccine and the lack of access to hospital facilities during the COVID-19 pandemic-related lockdown across the country. Second, the inclusion of subjects across all seasons to match the transmission dynamics of influenza disease in the region would have been useful for studying the effectiveness of the influenza vaccine during pregnancy. The data were collected from the obstetrics ward of the hospital, which is one of the selected hospitals offering the maternal influenza vaccine. Most of the women who deliver in this hospital attend their ANC visits in same hospital, so the comparatively moderately high influenza vaccine uptake reported in this study might not be representative of the region. Although the pregnancy outcome data were collected from women admitted in the obstetrics ward, some important data such as termination of pregnancies and cases of stillbirths might be missed, as those cases generally do not stay for a long duration in the hospital.

## 5. Conclusions

Our data provided reassuring evidence that an injectable inactivated trivalent influenza vaccine is safe during pregnancy. There was no increased risk of adverse birth outcomes, regardless of the trimester in which vaccination was performed. Despite an interrupted supply of influenza vaccine through the program, we observed moderately good coverage of maternal influenza vaccination among the population.

## Figures and Tables

**Figure 1 vaccines-11-01034-f001:**
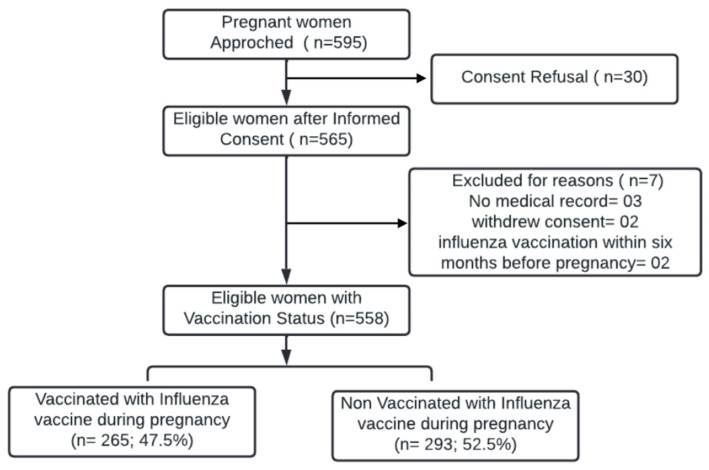
Flow diagram for subject disposition.

**Figure 2 vaccines-11-01034-f002:**
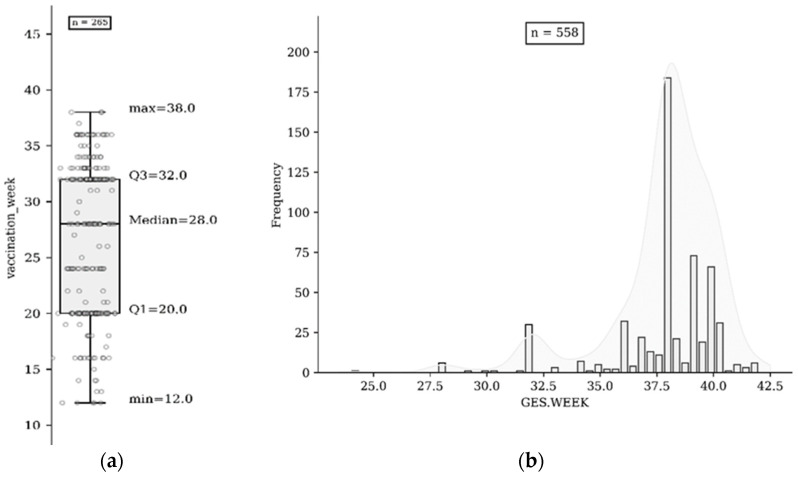
(**a**) Vaccination based on gestational age in weeks of participants (*n* = 265); (**b**) gestational weeks of participants (*n* = 558).

**Table 1 vaccines-11-01034-t001:** Maternal characteristics of vaccinated and unvaccinated pregnant women who delivered at an obstetric hospital in Pune between October 2019 and March 2020.

Parameters	Vaccinated (*n* = 265)	Unvaccinated (*n* = 293)	Total (*n* = 558)	*p*-Value
No. of participants	265 (47.5%)	293 (52.5%)	558	
**Maternal Age (years)**				
Age (yrs)	24.8 ± 4.5	24.7 ± 4.4	24.7 ± 4.4	
<18	2 (0.8%)	6 (2%)	8	*p* = 0.424
18 to 30	228 (86%)	251 (85.7%)	479	
>30	35 (13.2%)	36 (12.3%)	71	
**Parity**				
Primiparous	132 (49.8%)	129 (44%)	261	*p* = 0.171
Multiparous	133 (50.2%)	164 (56%)	297	
**Maternal Education**				
Illiterate	24 (9.1%)	84 (28.7%)	108	***p* < 0.001 ***
Primary	118 (44.5%)	121 (41.3%)	239	
Secondary	70 (26.4%)	60 (20.5%)	130	
Graduation	53 (20%)	27 (9.2%)	80	
Post-graduation	0 (0%)	1 (0.3%)	1	
**Smoking**				
No	262 (98.9%)	293 (100%)	555	0.106
Yes	3 (1.1%)	0	3	
**Alcohol**				
No	265 (100%)	293 (100%)	558	--
**Influenza vaccine at**	***n* (%)**	NA		
1st Trimester	5 (1.9%)			
2nd Trimester	107 (40.4%)			
3rd Trimester	153 (57.7%)			
**Mean (SD) gest. week of vaccine administration**	26.8 ± 6.9	NA	-	

The observations recorded in Table 1 shows that the distribution of maternal education differed significantly between groups of nonvaccinated and vaccinated participants (*p*-value < 0.05). A significantly higher proportion of vaccinated participants had a relatively higher level of education. * *p*-value < 0.05.

**Table 2 vaccines-11-01034-t002:** Delivery characteristics and pregnancy outcomes according to maternal influenza vaccination status with unadjusted (univariate) and adjusted (multivariable) odds ratios.

Delivery	Not Vaccinated (*n* = 293)	Vaccinated (*n* = 265)	Total (*n* = 558)	*p*-Value	Adjusted (Multivariable)		
Characteristics					Odds Ratio (AOR) #		
					AOR	95% CI	*p*-Value
**Gestational age**	37.6 ± 2.60	38.09 ± 2.19	37.83 ± 2.43	**0.02 ***			
Full-term	252 (86%)	228 (86%)	480 (86%)	**0.02 ***			
Post-term	8 (2.7%)	18 (6.8%)	26 (4.7%)		1	--	
Pre-term *	33 (11.3%)	19 (7.2%)	52 (9.3%)		1.69	0.92–2.99	0.14
**Birth weight**	2579 ± 602 g	2693 ± 503 g	2633 ± 560 g	**0.02 ***			
Normal	204 (69.6%)	191 (72.1%)	395 (70.8%)	0.12	1	--	
LBW	66 (22.5%)	64 (24.2%)	130 (23.3%)		0.86	0.74–1.54	0.61
VLBW	23 (7.8%)	10 (3.8%)	33 (5.9%)		2.29	1.03–5.58	**0.03 ***
**Mode of delivery**							
Normal	75 (25.6%)	66 (24.9%)	141 (25.3%)	0.85	1	--	
LSCS	218 (74.4%)	199 (75.1%)	417 (74.7%)		0.97	0.78–1.85	0.47
**Pregnancy outcome**							
Live birth	291 (99.3%)	264 (99.6%)	555 (99.5%)		1	--	
Stillbirth	2 (0.7%)	1 (0.4%)	3 (0.5%)	0.99	1.8	0.18–24.64	0.56
**Baby’s sex**							
Male	154 (52.6%)	144 (54.3%)	298 (53.4%)	0.67			
Female	139 (47.4%)	121 (45.7%)	260 (46.6%)		--		
**Gestational Hypertension**							
No	278 (94.9%)	257 (97.0%)	535 (95.9%)				
Yes	15 (5.1%)	8 (3.0%)	23 (4.1%)	0.22	--		
**Chorioamnionitis**							
No	291 (99.3%)	262 (98.9%)	553 (99.1%)				
Yes	2 (0.7%)	3 (1.1%)	5 (0.9%)	0.58	--		
**High-risk pregnancy**							
**No**	238 (81.2%)	218 (82.3%)	456 (81.7%)	0.75	1	--	--
**Yes**	55 (18.8%)	47 (17.7%)	102 (18.3%)		1.1	0.74–1.88	0.71
**Complications before**							
**or during delivery**							
**No**	238 (81.2%)	218 (82.3%)	456 (81.7%)	0.75	1	--	--
**Yes**	55 (18.8%)	47 (17.7%)	102 (18.3%)		1.09	0.72–1.74	0.7
**Complications after delivery**							
**No**	283 (96.6%)	261 (98.5%)	544 (97.5%)	0.18	1	--	--
**Yes**	10 (3.4%)	4 (1.5%)	14 (2.5%)		2.85	0.88–9.85	0.09

*—statistically significant (*p*-value < 0.05).

**Table 3 vaccines-11-01034-t003:** Birth outcomes following influenza vaccination in pregnancy at obstetric hospital with unadjusted (univariate) and adjusted (multivariable) odds ratios.

Variables	Not Vaccinated	Vaccinated	Total	*p*-Value	Adjusted (Multivariate)		
	(*n* = 293)	(*n* = 265)	(*n* = 558)		Odds Ratio (AOR)		
					AOR	95% CI	*p*-Value
**Apgar score (5 min)**							
≥7	289 (98.6%)	261 (98.5%)	550 (98.6%)	0.99	1	--	
<7	4 (1.4%)	4 (1.5%)	8 (1.4%)		0.92	0.23–3.78	0.6
**NICU required**							
Not required	288 (98.3%)	259 (97.7%)	547 (98%)	0.64	1	--	
Required	5 (1.7%)	6 (2.3%)	11 (2%)		0.87	0.29–2.85	0.55
**Mechanical ventilation**							
Not required	290 (99%)	261 (98.5%)	551 (98.7%)	0.71	1	--	
Required	3 (1%)	4 (1.5%)	7 (1.3%)		0.72	0.21–4.00	0.4
**Respiratory distress**							
No	287 (98%)	260 (98.1%)	547 (98%)	0.89	1	--	--
Yes	6 (2%)	5 (1.9%)	11 (2%)		1.1	0.35–3.95	0.72
**Fetal distress**							
No	282 (96.2%)	257 (97%)	539 (96.6%)	0.63	1	--	--
Yes	11 (3.8%)	8 (3%)	19 (3.4%)		0.29	0.51–3.87	0.4
**Congenital anomaly**							
No	291 (99.3%)	262 (98.9%)	553 (99.1%)	0.67	1	--	--
Yes	2 (0.7%)	3 (1.1%)	5 (0.9%)		0.81	0.10–3.87	0.47

*p*-values by chi-square test. Adjusted for maternal age, education, and gravidity using logistic regression analysis. *p*-value < 0.05 is considered to be statistically significant. Fetal distress: changes in heart rate, movement, or signs of oxygen deprivation in fetus before or during labor.

## Data Availability

Data are available upon request due to ethical restrictions and confidentiality. The data presented in this study are available upon request from the corresponding author. The data were not publicly available due to confidentiality and ethical restrictions.

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
