# Peer review of "Safety and Protective Effects of Influenza Vaccination in Pregnant Women on Pregnancy and Birth Outcomes in Pune, India: A Cross-Sectional Study"

_vaccines, 2023, doi:10.3390/vaccines11061034_

Round 1

Reviewer 1 Report

no critical comments

Author Response

The manuscript is revised to make improvement in introduction, methods, design, results and conclusion and references.

Reviewer 2 Report

Interesting article with a high social impact, well written but below are some changes to be made.

 1) Write acronyms extensively, at least the first time you use them;

 2) Explain well what do you mean for APGAR Score, LSCS, mode of delivering, list categorical variables. 

3) In paragraph 2.5 put the various steps of the statistical analysis in chronological order and, if possible, insert a scheme immediately afterwards. Insert a summary graph of the analytical process.

4) Figure 1 is more useful if moved before the paragraph 3.1, so as to give a graphical representation of what was written immediately before. 

At the AOU "Federico II" we also conducted a systematic review on vaccination coverage in the pediatric age group due to parental concern about the safety and efficacy of the various pediatric vaccines.

Author Response

1) Write acronyms extensively, at least the first time you use them;

Response:

Acronyms used for first time are explained (e.g. WHO, SAGE, TIV3, AOR, YCMH, ANC, PPROM, LBW…etc)

 2) Explain well what do you mean for APGAR Score, LSCS, mode of delivering, list categorical variables. 

Response:

The points elaborated in the manuscript as below…

APGAR score (a scoring system providing a standardized assessment for infants after delivery), A lower (uterine) segment Caesarean section (LSCS) mode of delivery,

Categorical variables: Age-groups, Sex, Parity, Education, Smoking

3) In paragraph 2.5 put the various steps of the statistical analysis in chronological order and, if possible, insert a scheme immediately afterwards. Insert a summary graph of the analytical process.

Response:

The statistical analysis was corrected in chronological order in text only.

4) Figure 1 is more useful if moved before the paragraph 3.1, so as to give a graphical representation of what was written immediately before.

 ‌ Response:

Updated as per suggestions.

Reviewer 3 Report

Thank you for the opportunity to review this interesting manuscript on Safety and Protective Effects of Maternal Influenza Vaccination on Pregnancy and  Birth Outcomes in pregnant women from Pune, India.

The research question that the authors set out to answer was to assess safety and protective effects of the influenza vaccine among pregnant women on pregnancy and birth outcomes. They performed a monocenter cross sectional study to answer that question. 

The study is well conducted. The manuscript is well written and easy to read. The focus is important and needs attention.

However, many modifications are needed.

Comments :

1.     Title: The authors may replace by: “Safety and Effects of Influenza Vaccination in pregnant women on Pregnancy and Birth Outcomes in Pune, India A Cross sectional Study »

2.     Abstract : Results: The protective effect of maternal influenza vaccination against delivering very low birth 25 weight infants was observed. (aOR 2.29, 95% CI 1.03 to 5.58, p=0.03). When conducting a cross-sectional study, the authors can conclude that there is an association, but there is no causal effect between Influenza vaccine and VLBW. The authors could write instead: “Reductions in VLBW associated with maternal immunization suggest possible protective effects

3.     Introduction: “Immunization of pregnant women with an influenza vaccine is effective in reducing the risk of influenza [1] and has been reported to be safe for mothers and their foetuses[2] » Please add the dots at the end of the sentences.

4.     It would be relevant to know the seasonality of influenza in India and especially in Pune and its area. Which months are usually concerned?

5.     A review of previous studies on the safety of influenza immunization among pregnant women revealed that no studies have examined the influence of influenza vaccination on adverse birth outcomes in pregnant Indian women. » I would suggest using the words “pregnancy and neonatal outcomes”.

6.     MM : « Intervention » should be replaced by « exposure » as it is not an interventional study, such as a randomized controlled trial.

7.     MM: “chorioamnionitis (premature preterm rupture of membranes or PPROM) » should be replaced by “chorioamnionitis, premature preterm rupture of membranes (PPROM),

« Multivariate logistic regression analysis was performed to obtain statistically significant and independent determinants of the incidence of abnormal outcome measures (such as delivery outcome and incidence of neonatal complications). » The authors should replace « multivariate” by “multivariable”, “was” by “were” .

While multivariable and multivariate regressions share similar functions and names, there is one key difference between them. In a multivariate regression, there are multiple independent variables and multiple outcomes. In multivariable regression, there are multiple independent variables, but only one outcome.

8.     « such as delivery outcome and incidence of neonatal complications ». Please replace by « pregnancy and neonatal outcomes”

9.     Results: It would be interesting to have information on the occurrence of influenza during pregnancy. It is not the primary objective of the study, but it should be at least a secondary objective.

10.   Line 149: “Though the number of spontaneous abortion cases were low » : please correct with “lower”.

11.   « (= no: <0.001, yes:0.001) ». Please clarify.

12.   It makes no sense to write “113.95 g ». First, the authors should write the median of the weights in each groups instead of this of the whole population. Second, please remove the decimals, because a birthweight is never precise at less than 1 gram!

13.   Please write: 47.5% (265/558) 

14.   3.1.2. Birth Weight variable : it is a neonatal outcome. In this paragraph, the description of the outcomes should be re-organized inside the paragraph and pregnancy outcomes should be described in the previous paragraph.

15.   Apgar score at 1’ has no clinical impact. The authors should omit this variable and keep only the Apgar score at 5’.

16.   The presentation of the results should be homogenous through the different parts (currently, there are descriptive findings, then p-values, then crude OR)

17.   Figure 1: arrows directed to the right to show the excluded women would make the flowchart clearer.

18.   Table 1: You can remove “NS”. It is obvious.

19.   Table 1: Please define the different preterm categories in the legend.

20.   Regarding the text below table 1, it is not clear whether it is a legend or the text itself.

21.   Table 2: What does LSCS mean? Please add the meaning in the legend. How were selected the confounding factors of the multivariable logistic regression? 

22.   Confounding. Have the authors considered all important confounders and controlled for them in their analysis? Could there be residual confounding by variables that have not been considered or because of incomplete adjustment for factors that have? The information on medical history, such as cardiovascular, pulmonary pre-exiting disease, pre-existing diabetes or obesity is critical as flu vaccination is recommended to these persons in general population. Medical history can be a confounding factor, as these conditions are associated with both influenza vaccine uptake and low birthweight. If the authors can not get the medical information, this potential confounding bias should be addressed in the discussion.

23.   Table 3: Please define “foetal distress”, “high-risk pregnancy”. Please reorder the outcomes: pregnancy outcomes first and then neonatal outcomes (chronological order).

24.   Figure 2: Please replace “number of gestation. Weeks” by “gestational age in weeks”

25.   If the minimal gestational age was 12 weeks of gestation, the authors must precise that spontaneous abortion is between 12 and 20 weeks.

26.   Discussion: The finding of lower VLBW in the unvaccinated group may be explained by chance, since there are multiple comparisons in the univariate analyses and multivariable logistic regressions.

27.   « We did not find any association between spontaneous abortion and inactivated influenza vaccination during pregnancy. » It is normal since the earliest flu vaccination was performed at 12 weeks.

28.   How did the COVID-19 pandemic affect the results? When was the lockdown in India? Did the pregnant women wear face masks in India? If so, lockdown and mask may have decreased the risk of getting influenza during the pandemic period (at the end of the study period).

Would it be possible to consider the calendar period related to the pandemic /lockdown) in the analyses? For instance, comparison of period 1=October 19 – February 2020 with Period 2=March 2020? Or the dates of the lockdown in India?

28. “We also observed that vaccine uptake for influenza vaccination among pregnant women is good in the Pune area, provided that there is an uninterrupted supply of vaccine. » Please rephrase this sentence.

29. « The findings of this study underscore the importance of maternal influenza vaccination in preventing influenza in 296 both pregnant women and their infants. « 

The authors may not conclude this, as influenza events were not compared between both groups (of women and their children). This sentence is not related to the current study.

30. Selection bias:

-        Do the woman who have miscarriage usually present to maternity ward or to their OBGYNs in Pune area? The authors should explain the follow-up of pregnant women more in details in order to demonstrate whether there is a risk of selection bias.

-        Is this hospital a tertiary center, with high-risk pregnancies and a neonatal ICU? It would be relevant to describe this in the methods section and in the discussion if there is a potential selection bias.

Round 2

Reviewer 3 Report

I am happy with the responses of the authors.

However, it is too complex to read the revised manuscript with a clean copy only. Is it possible to get a highlighted revised version or a tracking mode version?

Therefore, I only looked at the figures and tables and found some corrections to do:

"3.1.1. Pregnancy outcomes" is missing

3.1.2 Neonatal Outcomes is better with S as there are several outcomes

Add n=595 in the flowchart

In the tables, COR should be defined in the legend.

Multivariate has not been changed by multivariable, as suggested.  In a multivariate regression, there are multiple independent variables and multiple outcomes. In multivariable regression, there are multiple independent variables, but only one outcome.

Table 3: Please remove all the "ns"

I could not find the definition of "high risk pregnancy" and "fetal distress".

I need a highlighted version to check the changes in the manuscript.

Author Response

I am happy with the responses of the authors.

However, it is too complex to read the revised manuscript with a clean copy only. Is it possible to get a highlighted revised version or a tracking mode version?

Response: Both clean and TC version of revised manuscript including round 2 comments are uploaded

Therefore, I only looked at the figures and tables and found some corrections to do:

"3.1.1. Pregnancy outcomes" is missing

Response:   Section “3.1.1. Pregnancy outcomes” is restored in the revised manuscript.

3.1.2 Neonatal Outcomes is better with S as there are several outcomes

Response:   Neonatal outcomes is corrected in the revised manuscript.

Add n=595 in the flowchart

Response:  n= 595 is added in flochart

In the tables, COR should be defined in the legend.

Response:   column with COR is deleted and Adjusted Odds Ratio (AOR) is retained

Multivariate has not been changed by multivariable, as suggested.  In a multivariate regression, there are multiple independent variables and multiple outcomes. In multivariable regression, there are multiple independent variables, but only one outcome.

Response:   Multivariate has been replaced by Multivariable.

Table 3: Please remove all the "ns"

Response:   word “NS” has been removed from the tables.

I could not find the definition of "high risk pregnancy" and "fetal distress".

Response:   High Risk pregnancy and fetal distress is defined in legends.

I need a highlighted version to check the changes in the manuscript.

Response:   Both clean and TC version of revised manuscript including round 2 comments are uploaded

Round 3

Reviewer 3 Report

Please find minor comments:

Abstract : line 25 : Please replace “Reductions in VLBW were observed which could be associated with maternal immunization » by “Women not vaccinated against influenza during pregnancy had higher risk of delivering very LBW infants » to fit with the direction of aOR.

Lines 176-177 : Please write “Apgar” (and line 304) and remove “( a scoring system providing a standardized assessment for infants after delivery) ». It is very well known by the perinatal healthcare providers.

Line 178: Please write “neonatal care unit hospitalization (NICU)”

Line 237: participants

Line 309: Please chose between g and gm

Line 310: If the distribution was not normal, please use median and IQR instead of mean and SD.

Line 331: Please replace “Maternal influenza vaccination was protective against delivering Very LBW term infants » by “Women not vaccinated against influenza during pregnancy had higher risk of delivering very LBW infants » to fit with the aOR.

Line 332: adjusted odds ratio [AOR] : this should be placed the first time aOR was used.

Line 338: Please remove “and vice versa (p-value<0.05). »

Line 421-425: Please remove: “Table 3 shows that the distribution of the incidence of various neonatal complications such as Abnormal APGAR score (at 5-min), NICU requirement, requirement of mechanical ventilation, occurrence of respiratory distress, occurrence of fetal distress, occurrence of congenital anomaly, were also not significantly associated with vaccination status (P-value>0.05).”

The following paragraph says exactly the same.

Line 434: showed

Line 457: Please change for "may be explained just by chance"   

Line 470: Please remove “The first trimester is a significant period for embryogenesis of major organs, and pregnant women are hesitant to vaccinate during this period. The results of this study suggest no adverse effects on birth outcomes, even if influenza vaccination is administered during the first trimester of pregnancy.” Since there are only 5cases vaccinated during first trimester.

Line 515: Remove “and limitations. ».

Line 530 « also » may be replaced by “although”?

Could be improved

Author Response

Abstract: line 25 : Please replace “Reductions in VLBW were observed which could be associated with maternal immunization » by “Women not vaccinated against influenza during pregnancy had higher risk of delivering very LBW infants » to fit with the direction of aOR.

Response:  Corrected

Lines 176-177 : Please write “Apgar” (and line 304) and remove “( a scoring system providing a standardized assessment for infants after delivery) ». It is very well known by the perinatal healthcare providers.

Response:  Corrected- to “Apgar” and “( a scoring system providing a standardized assessment for infants after delivery” deleted.

Line 178: Please write “neonatal care unit hospitalization (NICU)”

Response:  Corrected

Line 237: participants

Response:  Corrected

Line 309: Please chose between g and gm

Response:  Corrected, (gm)

Line 310: If the distribution was not normal, please use median and IQR instead of mean and SD.

Response: A line “Where distribution was not normal median and IQR were used” added and for the Variables like Age, gestational age and Birth weight, the data is in normal distribution (as SD is less than half of the mean in all three variables). Hence no need for Median & IQR but of Mean & SD (as we mentioned in all 3 tables). 

Line 331: Please replace “Maternal influenza vaccination was protective against delivering Very LBW term infants » by “Women not vaccinated against influenza during pregnancy had higher risk of delivering very LBW infants » to fit with the aOR.

Response:  Corrected

Line 332: adjusted odds ratio [AOR] : this should be placed the first time aOR was used.

Response: adjusted odds ratio [AOR] Added in Results section where used for first time and “adjusted odds ratio” deleted wherever it was repeated.

Line 338: Please remove “and vice versa (p-value<0.05). »

Response:  The words are deleted

Line 421-425: Please remove: “Table 3 shows that the distribution of the incidence of various neonatal complications such as Abnormal APGAR score (at 5-min), NICU requirement, requirement of mechanical ventilation, occurrence of respiratory distress, occurrence of fetal distress, occurrence of congenital anomaly, were also not significantly associated with vaccination status (P-value>0.05).”

Response: Lines deleted

The following paragraph says exactly the same.

Line 434: showed

Response:  Corrected

Line 457: Please change for "may be explained just by chance"   

Response:  Corrected

Line 470: Please remove “The first trimester is a significant period for embryogenesis of major organs, and pregnant women are hesitant to vaccinate during this period. The results of this study suggest no adverse effects on birth outcomes, even if influenza vaccination is administered during the first trimester of pregnancy.” Since there are only 5cases vaccinated during first trimester.

Response:  Corrected

Line 515: Remove “and limitations. ».

Response:  Words “and Limitations are deleted”

Line 530 « also » may be replaced by “although”?

Response:  Corrected